# Flt3 Activation Mitigates Mitochondrial Fragmentation and Heart Dysfunction through Rebalanced L-OPA1 Processing by Hindering the Interaction between Acetylated p53 and PHB2 in Cardiac Remodeling

**DOI:** 10.3390/antiox12091657

**Published:** 2023-08-22

**Authors:** Kaina Zhang, Yeqing Zheng, Gaowa Bao, Wenzhuo Ma, Bing Han, Hongwen Shi, Zhenghang Zhao

**Affiliations:** 1Department of Pharmacology, School of Basic Medical Sciences, Xi’an Jiaotong University Health Science Center, Xi’an 710061, China; 2Institute of Cardiovascular Sciences, Translational Medicine Institute, Xi’an Jiaotong University Health Science Center, Xi’an 710061, China

**Keywords:** cardiac remodeling, mitochondrial dynamics imbalance, prohibitins, p53, FMS-like receptor tyrosine kinase 3, OPA1

## Abstract

Recent studies have shown that FMS-like receptor tyrosine kinase 3 (Flt3) has a beneficial effect on cardiac maladaptive remodeling. However, the role and mechanism of Flt3 in mitochondrial dynamic imbalance under cardiac stress remains poorly understood. This study aims to investigate how Flt3 regulates p53-mediated optic atrophy 1 (OPA1) processing and mitochondrial fragmentation to improve cardiac remodeling. Mitochondrial fragmentation in cardiomyocytes was induced by isoprenaline (ISO) and H_2_O_2_ challenge, respectively, in vitro. Cardiac remodeling in mice was established by ligating the left anterior descending coronary artery or by chronic ISO challenge, respectively, in vivo. Our results demonstrated that the protein expression of acetylated-p53 (ac-p53) in mitochondria was significantly increased under cell stress conditions, facilitating the dissociation of PHB2-OPA1 complex by binding to prohibitin 2 (PHB2), a molecular chaperone that stabilizes OPA1 in mitochondria. This led to the degradation of the long isoform of OPA1 (L-OPA1) that facilitates mitochondrial fusion and resultant mitochondrial network fragmentation. This effect was abolished by a p53 K371R mutant that failed to bind to PHB2 and impeded the formation of the ac-p53-PHB2 complex. The activation of Flt3 significantly reduced ac-p53 expression in mitochondria via SIRT1, thereby hindering the formation of the ac-p53-PHB2 complex and potentiating the stability of the PHB2-OPA1 complex. This ultimately inhibits L-OPA1 processing and leads to the balancing of mitochondrial dynamics. These findings highlight a novel mechanism by which Flt3 activation mitigates mitochondrial fragmentation and dysfunction through the reduction of L-OPA1 processing by dampening the interaction between ac-p53 and PHB2 in cardiac maladaptive remodeling.

## 1. Introduction

Cardiac remodeling refers to significant changes in the structure of cardiac tissue, including changes in ventricular mass, volume or geometry, fibrotic content, vascularization, cardiac muscle cell hypertrophy, and apoptosis [1]. Typically initiated by pathological cardiomyocyte hypertrophy, this process leads to structural heterogeneity of the myocardium, along with myocardial fibrosis and deleterious structural remodeling [2]. It also occurs post-myocardial infarction, involving cardiomyocyte necrosis and apoptosis. Surviving cells undergo hypertrophy and hyperplasia in response to nutrient and oxygen deprivation [3]. Mitochondria, serving as the central hub for coordinating energy transduction in cardiomyocytes, constantly undergo fission and fusion to form a dynamic network. Mitochondrial dysfunction is commonly attributed to the disruption of cardiomyocyte architecture, which in turn is associated with cardiac hypertrophy and ischemia/reperfusion injury [1]. Therefore, it is imperative to delineate a novel therapeutic strategy that targets mitochondrial dynamic imbalance to improve cardiac remodeling.

Emerging evidence indicates that the proteolytic processing of OPA1, a dynamin-like GTPase localized in the inner mitochondrial membrane, plays a crucial role in maintaining proper mitochondrial dynamics. OPA1 can be cleaved into long and short forms (L-OPA1, S-OPA1) by two mitochondrial proteases, namely OMA1 and AAA protease YMEIL. Of note, L-OPA1 primarily facilitates mitochondrial fusion, while excessive S-OPA1 accelerates mitochondrial fission [4,5,6,7], resulting in an imbalance between L-OPA1 and S-OPA1 that triggers mitochondrial fragmentation and apoptosis. 

Recently, novel evidence suggests that the cleavage of OPA1 is regulated by the prohibitin complex, a heteromeric ring-shaped scaffolding complex located in the inner mitochondrial membrane and composed of prohibitin 1 (PHB1) and prohibitin 2 (PHB2) [8]. This complex has been reported to play a pivotal role in maintaining both mitochondrial metabolism and structure [9,10]. PHB1, located in the nucleus, cytoplasm, and mitochondria, while PHB2, predominantly localized within the mitochondria, plays a critical role in regulating mitochondrial dynamics [10]. Lately, Li et al. revealed that liver PHB2 deficiency leads to excessive proteolytic cleavage of L-OPA1, resulting in mitochondrial fragmentation and increased apoptosis [11]. However, it remains unclear whether PHB2 or PHB1 proteins act alone or in conjunction with other proteins, such as p53, to regulate L-OPA1 processing and mitochondrial dynamics during cardiac remodeling. Receptor tyrosine kinase 3 (Flt3), a member of the class III receptor tyrosine kinase family, is activated by a specific Flt3 ligand (FL) and leads to rapid autophosphorylation of the receptor. Activation of Flt3 induces the activation of many intermediate signal-transduction mediators, determining cell survival, proliferation, differentiation, and metabolism [12]. Recent studies from ours [13,14] and others [15] demonstrated a notable augmentation in the expression of Flt3 in hearts subjected to pathological stress, but its physiological functions and pathological significance remain largely unknown. Our previous studies have demonstrated that cardiac pathological stress triggers the expression of Flt3 in cardiomyocytes, and Flt3 activation with its specific ligand (FL) remarkably reversed Angiotensin II or isoprenaline (ISO)-induced cardiac hypertrophy and cardiac adverse remodeling through AMPK/mTORC1/FoxO3a or SIRT1/p53 signaling [13,14]. However, the mechanism by which Flt3 regulates L-OPA1 processing is not elaborated. Our novel findings in the present study demonstrated that acetylated-p53 translocated to the inner mitochondrial membrane and replaced OPA1 from the OPA1-PHB2 complex, forming an ac-p53-PHB2 complex. This promoted L-OPA1 processing and consequently increased mitochondrial fragmentation, contributing to both ISO- and LAD (ligating the left anterior descending coronary artery)-induced cardiac remodeling. Chronic ISO insult or acute myocardial ischemia infraction by LAD ligation is the most common inducer for cardiac remodeling in animals such as mice or rats, which involves distinct heart pathogenesis. ISO can activate β receptors and their downstream signaling pathways in cardiomyocytes. Hence, chronic ISO administration is extensively used to simulate cardiac hypertrophy and remodeling as well as heart failure in animals via chronic overactivity of sympathetic nerves, such as hypertension-induced cardiac remodeling. The acute ischemia and/or reperfusion triggers a burst of ROS products, leading to Ca^2+^ overload, apoptosis, and necrosis of cardiomyocytes. Therefore, ISO-induced cardiac remodeling is characterized by cardiomyocyte hypertrophy due to persistent sympathetic hyperactivity along with ROS generation, apoptosis, and metabolic disturbance, including mitochondrial dysfunction [16]. In contrast, LAD-induced cardiac remodeling involves cardiomyocyte apoptosis, necrosis, the disorder of mitochondrial morphology and function, as well as the loss in the number of cardiomyocytes [17]. Nevertheless, all of these insults elicit mitochondrial dynamics imbalance and oxidative stress while reducing the expression of SIRT1 [13]. Therefore, we conducted these two experimental models to investigate how Flt3 regulates p53-mediated L-OPA1 processing in the cardiomyocytes. Flt3 activation led to significant attenuation in L-OPA1 processing by inhibiting the interaction between acetylated-p53 and PHB2 within mitochondria, resulting in decreased mitochondrial fragmentation. This ultimately improved cardiac hypertrophy and remodeling. This study aims to investigate whether the regulation of L-OPA1 processing by the ac-p53-PHB2 complex in cardiomyocytes represents a universal mechanism underlying mitochondrial dynamics imbalance in ISO- and LAD-induced cardiac remodeling, and it further explores the potential of Flt3 as a target for improving mitochondrial dynamics in this context.

## 2. Materials and Methods

### 2.1. Animal Experiment Protocol

Ten-week-old male C57BL/6 mice (25–28 g) were procured from Xi’an Jiaotong University Laboratorial Animal Center. Mice were routinely fed and housed as previously described [13]. 

The myocardial infarction (MI) model was established in mice by permanent ligation of the left anterior descending (LAD) coronary artery under isoflurane anesthesia (2% inhalation). Briefly, following the ligation of the LAD coronary artery in mice using a 6.0 silk suture positioned 2.0 mm inferior to the left auricle, standard lead II electrocardiogram (ECG) signals were recorded by a polygraph recorder BL-420S system (Taimeng, Chengdu, China). A successful MI model was confirmed by significant ST-segment elevation. Mice with left ventricular ejection fraction ranging from 40% to 50% three days post-surgery were included in the study and randomly allocated into two groups: one group received intraperitoneal administration of Flt3-ligand (FL, 5 μg/kg, Prospec, Ness-Ziona, Israel) every other day for four weeks (MI + FL group), while the other group was administered normal saline (MI group). Sham-operated mice without ligation served as controls, received normal saline (Sham group), and were treated with FL at a similar administration (Sham + FL). After treatment completion, the mice were anesthetized for heart function measurements and extracted hearts for histomorphology. The left ventricular tissues of the infarct border zone were collected for western blotting analysis of the associated proteins. 

The model of cardiac hypertrophy and remodeling was established in mice by subcutaneous injection of isoprenaline (ISO, 6 mg/kg, I8480, Solarbio, Beijing, China) once a day for 7 consecutive days, while FL was intraperitoneally administered 2 h before ISO every day as previously described [13].

### 2.2. Isolation, Culture, and Treatment of NRCMs

Neonatal rat cardiomyocytes (NRCMs) were isolated from 3-day-old Sprague Dawley rats, as previously described [13]. Briefly, neonatal rat hearts were minced in PBS and then digested 6 times for 5 min each with a refreshed enzyme dilution consisting of 0.1% trypsin (Amersco, WA, USA) and 0.1% collagenase type I (1 mg/mL, 125 U/mg, Worthington, OH, USA). After cardiac fibroblasts were removed, cardiomyocytes were cultured in DMEM/low-glucose Medium (Gibco, Waltham, MA, USA) supplemented with FBS (Gibco, Grand Island, NY, USA), as well as penicillin and streptomycin (Beyotime Biotechnology, Shanghai, China) at 37 °C in 5% CO_2_ and 95% air for 48–72 h. The cells were starved in serum-free medium for 12 h, and then were treated with different interventions, including the following groups: (i) CON, the cells were maintained in serum-free medium with the vehicle; (ii) ISO, the cells were treated with 10 μM ISO for 24 h; (iii) ISO + FL, the cells were exposed to 200 ng/mL FL for 2 h prior to treatment with 10 μM ISO for 24 h; (iv) FL, the cells were treated with 200 ng/mL FL alone for 6 or 24 h; (v) ISO + pifithrin-α (PFT-α, HY-123076, MCE, NJ, USA), the cells were exposed to 20 μM PFT-α for 1 h followed by treatment with 10 μM ISO for 24 h; (vi) ISO + FL + EX527 (HY-15452, MCE, NJ, USA), the cells pretreated with 20 μM EX527 for 30 min were exposed to 200 ng/mL FL for 2 h followed by treatment with 10 μM ISO for 24 h; (vii) H_2_O_2_, the cells were treated with 200 μM H_2_O_2_ (Sigma-Aldrich) for 6 h; (viii) H_2_O_2_ + FL, the cells were exposed to 200 ng/mL FL for 2 h prior to treatment with 200 μM H_2_O_2_ for 6 h; (ix) H_2_O_2_+ PFT-α, the cells were exposed to 20 μM PFT-α for 1 h followed by treatment with 200 μM H_2_O_2_ for 6 h; (x) H_2_O_2_ + FL + EX527, the cells pretreated with 20 μM EX527 for 30 min were exposed to 200 ng/mL FL for 2 h followed by treatment with 200 μM H_2_O_2_ for 6 h; (xi) NC, after the cells were incubated with negative control siRNA for 4 h, the mixture of transfection was replaced with normal DMEM; (xii) siOPA1, the cells were incubated with siOPA1 (GenePharma, Shanghai, China) for 4 h, and then cultured in normal DMEM; (xiii) siOPA1 + H_2_O_2_, after exposure to siOPA1 for 4 h, the cells were incubated with normal DMEM for 20 h, and then were exposed to 200 μM H_2_O_2_ for 6 h; (xiv) siOPA1 + H_2_O_2_ + FL, after being exposed to siOPA1 for 4 h, the cells were cultured with DMEM. 24 h later, they were exposed to 200 ng/mL FL for 2 h, followed by 200 μM H_2_O_2_ for 6 h.

### 2.3. Echocardiography

Four weeks post-surgery, the mice were anesthetized with 1–2% isoflurane inhalation to obtain cardiac function via echocardiography using a 15 MHz linear transducer (VEVO 2100; VisualSonics, Toronto, ON, Canada). The system software (Vevo LAB 3.1.1) package was utilized for data analysis to obtain parameters including left ventricular internal diastolic (LVIDd) and systolic (LVIDs) diameters, left ventricular ejection fraction (LVEF%), left ventricular fractional shortening (LVFS%), as well as left ventricular end-diastolic (LVEDV) and systolic volumes (LVESV).

### 2.4. Electrocardiographic Recordings

Standard lead II electrocardiogram (ECG) signals were recorded by a polygraph recorder BL-420S system (Taimeng, Chengdu, China) in anesthetized mice. The signals were calibrated to 1 mV/10 mm for each mouse in all groups. 

### 2.5. Histomorphology, Masson’s Trichrome Staining

Mouse hearts were excised, fixed with 4% paraformaldehyde for 24 h, embedded in paraffin, and sliced into sections of 4–5 μm thickness. Hematoxylin-Eosin and Masson’s trichrome staining were performed on the sections as previously described [13]. The cardiac sections were observed under a digital microscope (IX53; Olympus, Tokyo, Japan) and analyzed using ImageJ 1.44p software. 

### 2.6. Immunofluorescence Staining

Following the various treatments, NRCMs were fixed in 4% paraformaldehyde for 10 min and subsequently permeabilized with 0.3% Triton X-100 for 5 min. Following by washing 3 times with PBS, the NRCMs were blocked in 5% BSA for 30 min and then incubated with OPA1 antibody (1:180, 66853-1-Ig, Proteintech, Wuhan, China), and PHB2 antibody (1:50, 12295-1-AP, Proteintech, Wuhan, China) mixture overnight at 4 °C. Secondary antibodies conjugated with Goat Anti-Rabbit IgG (H + L) Fluor594 (1:50, S0006, Affinity, Liyang, China) and Goat Anti-Mouse IgG (H + L) Fluor488 (1:50, S0017, Affinity, Liyang, China) were incubated at room temperature for 1.5 h. Then nuclei were stained with DAPI staining solution for 10 min in the dark. Images were acquired with Nikon A1 confocal microscope (Nikon, New York, NY, USA) using ×63 oil immersion objective. The colocalization analysis of OPA1 and PHB2 was performed based on Spearman’s correlation coefficient using Image J Coloc2 software.

### 2.7. Annexin V-FITC/PI Apoptosis Assay 

The Annexin V-FITC/PI apoptosis detection kit (KGA107, KeyGEN BioTECH, Nanjing, China) was used to quantify the level of apoptosis in NRCMs. Following treatments as indicated, the NRCMs were washed with pre-cold PBS, digested using 0.25% trypsin without EDTA, and then resuspended in 1 × Binding Buffer. The sample was incubated with appropriate Annexin V-FITC for 15 min in the dark. Subsequently, PI was added and incubated on ice for 5 min. Finally, the cells were resuspended in 1 × Binding Buffer and smeared for fluorescence analysis. The images were acquired using a fluorescence microscope. Ten random photographs were taken from each group, and at least 500 individual cells were examined in each group.

### 2.8. Measurement of ROS

DCFH-DA probe (S0033S, Beyotime, Shanghai, China) was used for the detection of intracellular ROS. After the indicated treatments, the NRCMs were washed twice with PBS and subsequently incubated in serum-free DMEM with 10 µM DCFH-DA at 37 °C in the dark for 30 min. The cells were then washed 3 times with serum-free DMEM, and images were captured using a fluorescence microscope at 200× magnification. Ten random photographs were taken from each group in three independent experiments, and the mean fluorescence intensity of ROS was quantified using ImageJ 1.41o software based on the observation of at least 300 individual cells in each group.

### 2.9. Mitochondrial Tracking

After agent treatment, the NRCMs were washed twice with PBS, followed by incubating with 200 nM Mito-Tracker Green (Beyotime, Shanghai, China) solution at 37 °C in the dark for 45 min. Images were acquired with a fluorescence microscope. The images of mitochondria were photographed, and the morphological characteristics of mitochondria in different groups were analyzed as previously described [13]. 

### 2.10. Quantitative Real-Time PCR (RT-qPCR)

Trizol reagent (Genstar, Beijing, China) was utilized for the extraction of total RNA from NRCMs. PrimeScript RT reagent kit (Genstar, Beijing, China) was employed to synthesize cDNAs. The primers used in this study are shown in Table 1. PCR amplifications were quantified using the 2 × RealStar Green Fast Mixture (Genstar, Beijing, China) in the CFX Connect™ Real-Time System (Bio-RAD). The relative expression of mRNA was measured using the 2^−△△ct^ method.

### 2.11. Western Blotting Analysis

Protein extraction and western blotting analysis were performed as previously described [13]. The proteins of cardiac tissues and NRCMs were extracted using cell lysis buffer for Western and IP (P0013, Beyotime, Shanghai, China), supplemented with a protease inhibitor cocktail (Roche, Mannheim, Germany). The protein concentration was determined using the BCA assay kit (Beyotime, Shanghai, China). Equal amounts of protein (20–40 μg) were separated by 10% SDS-PAGE and transferred onto a PVDF membrane (Millipore, Boston, MA, USA). The primary antibodies, including anti-p53 (1:1000, 2524, Cell signaling technology, Danvers, MA, USA), anti-SIRT1 (1:1000, 9475, Cell signaling technology, Danvers, MA, USA), anti-ac-p53 (1:1000, AF4363, Affinity, Liyang, China), anti-OPA1 (1:1000, 66853-1-Ig, Proteintech, Wuhan, China), anti-PHB2 (1:2000, 11242-1-AP, Proteintech, Wuhan, China), anti-PHB1 (1:1000, 10787-1-AP, Proteintech, Wuhan, China), anti-COXIV (1:5000, 10787-1-AP, Proteintech, Wuhan, China) and anti-Beta actin (1:5000, 66009-1-Ig, Proteintech, Wuhan, China) were incubated overnight at 4 °C. Secondary antibody conjugated with horseradish peroxidase, including Goat Anti-Mouse IgG (1:5000, AB0102, Abways, Beijing, China), Goat Anti-Rabbit IgG (1:5000, AB0101, Abways, Beijing, China) or IPKine™ HRP, Mouse Anti-Rabbit IgG LCS (1:2000, A25022, Proteintech, Wuhan, China) was incubated for 1.5 h. Beta-actin was used as a protein loading control. The blots were developed using ECL reagent SuperKine™ West Pico PLUS Chemiluminescent Substrate (Abbkin, Wuhan, China) and visualized by a gel imaging system (SYNGENE, G: BOX). Protein expression levels were quantified by using Image J.

### 2.12. Transient cDNA Transfection

A wild-type p53 plasmid and p53 mutant constructs (K371R) plasmid of rat were constructed by GenePharma Corporation (Shanghai, China) using pcDNA3.1(+) vector. The wild-type p53 was site-specifically mutated according to the manufacturer’s instructions. Since the NRCMs are difficult to be transfected, H9c2 rat embryonic cardiac myoblasts were transfected with p53 and p53-K371R cDNA (1 μg/well, 48 h) in 6-well culture plates using TransIT-X2^®^ Dynamic Delivery System. The empty vector was used as a control, and following transfection, the cells were subjected to either 10 μM ISO for 24 h or 200 μM H_2_O_2_ for 6 h for immunoprecipitation or western blotting analysis.

### 2.13. Immunoprecipitation (IP)

Mitochondrial fractions were obtained by differential centrifugation. The myocardium of infarct border zone tissues or NRCMs was lysed with mitochondrial isolation buffer and subsequently resuspended in homogenizing buffer containing a protease inhibitor cocktail (Roche, Mannheim, Germany). After 60 strokes of homogenization in a Dounce homogenizer, the sample was centrifuged at 2000 g for 10 min at 4 °C to pellet the nuclear fraction. The supernatant was then removed and further centrifuged at 14,000× *g* for 20 min at 4 °C to pellet the mitochondrial fraction for subsequent immunoprecipitation.

One milligram of protein sample was incubated with either PHB2 Rabbit Polyclonal antibody (2 μg, 11242-1-AP, Proteintech, Wuhan, China) or PHB1 Rabbit Polyclonal antibody (2 μg, 10787-1-AP, Proteintech, Wuhan, China) at 4 °C overnight. The resulting mixture was then incubated with Protein A Agarose (Fast Flow for IP, P2051, Beyotime, Shanghai, China) for 2 h at 4 °C. After pelleting the agarose beads and resuspending them in sample buffer, the mixture was boiled and loaded onto a 10% SDS-PAGE gel. The related protein contents were examined by western blotting.

### 2.14. Statistical Analysis

All data are represented as the mean ± SD, and statistical analysis was performed using GraphPad Prism 8.0 software (San Diego, CA, USA). Statistical significance was performed using one-way ANOVA followed by Tukey’s post-hoc test, with unpaired Student’s *t*-test. Values of *p* less than 0.05 were considered statistically significant.

## 3. Results

### 3.1. Flt3 Activation Restored ISO- or H_2_O_2_-Induced Mitochondrial Dynamics Imbalance by Reducing L-OPA1 Processing 

It has been well documented that OPA1 is regulated by proteolytic cleavage, which leads to the balanced products of noncleaved, L-OPA1 and cleaved, S-OPA1 forms. Functional L-OPA1 is sufficient to promote mitochondrial fusion and maintain normal cristae in cardiomyocytes, while S-OPA1 appears to function in mitochondrial fission [5]. Our previously published study has demonstrated that Flt3 activation alleviates mitochondrial dynamic balance in cardiomyocytes stimulated by ISO in vitro and in vivo, as evidenced by increased mitochondrial fusion markers, OPA1 and Mfn2, as well as decreased mitochondrial fission marker Drp1, which occurs in a SIRT1/p53 dependent manner [13]. Here, we further found that Flt3 activation impeded mitochondrial L-OPA1 processing in ISO or H_2_O_2_ challenged-cardiomyocytes. This phenomenon is attributed to the enhanced stabilization of L-OPA1 with PHBs by reducing ac-p53 levels in mitochondria. Firstly, immunoblotting analysis revealed that ISO or H_2_O_2_ treatment significantly decreased the protein expression of mitochondrial L-OPA1, PHB1, and PHB2 in NRCMs, along with increased ac-p53 levels in mitochondria compared to the control group (Figure 1A,B). In contrast, FL intervention was able to reverse these protein changes induced by ISO or H_2_O_2_ in NRCMs. Subsequently, we further investigated the involvement of the SIRT1/p53 pathway in regulating L-OPA1 processing. As depicted in Figure 2A,B, our results showed that ISO or H_2_O_2_ treatment significantly reduced the expression of total L-OPA1, SIRT1, PHB1, and PHB2 proteins while increasing total p53 and mitochondrial ac-p53 levels compared to control cells. These alterations were greatly reversed by the application of FL or p53 inhibitor PFT-α. Furthermore, the impacts of FL intervention on ISO or H_2_O_2_ were completely nullified by SIRT1 inhibitor EX527. More importantly, we observed an increase in protein levels of ac-p53 in mitochondria accompanied by L-OPA1 processing. These results suggest that PHBs and ac-p53 in mitochondria probably regulate the L-OPA1 processing in a SIRT1/p53-dependent manner in cardiomyocytes responding to ISO or H_2_O_2_ stress. Morphologically, visualization with Mito-Tracker staining further displayed that FL remarkably reduced punctate mitochondria and increased the number of mitochondrial networks, which were replicated by PFT-α but abrogated by EX527 in response to H_2_O_2_ challenge seen in Figure 3 or ISO stimulation [13]. 

### 3.2. Flt3 Activation Inhibited H_2_O_2_-Induced ROS and Apoptosis by Improving Mitochondrial Dynamics Disturbance In Vitro 

Our previous work has confirmed that activating Flt3 efficiently suppresses cardiomyocyte hypertrophy by improving mitochondrial dynamics through the SIRT1/p53 pathway in the setting of ISO stimulation in vitro and in vivo [13]. Herein, we further investigated the impact of FL improving mitochondrial dynamics on H_2_O_2_-induced ROS and apoptosis. As shown in Figure 4, treatment with H_2_O_2_ significantly increased ROS and apoptosis levels, as evidenced by the elevated DCF fluorescence and FITC/PI-positive cells. Unsurprisingly, pretreatment with FL or PFT-α significantly ameliorated H_2_O_2_-induced oxidative stress and apoptosis, while EX527 pretreatment reversed the protective effects of FL intervention on ROS and apoptosis induced by H_2_O_2._ Of note, we knocked down OPA1 with siOPA1 and observed that H_2_O_2_ treatment significantly induced mitochondrial fragmentation, as well as enhanced ROS and apoptosis levels in the NRCMs. However, FL treatment could not attenuate H_2_O_2_-induced ROS and apoptosis. These results indicate that activating Flt3 attenuates cardiac remodeling in response to ISO or H_2_O_2_ insults, possibly by improving mitochondrial dynamics disturbance through SIRT1/p53 pathway.

### 3.3. Interaction between ac-p53 and PHBs Contributed to Mitochondrial L-OPA1 Processing, Flt3 Activation Attenuated This Interaction, thus Increasing L-OPA1 and Improving ISO- or H_2_O_2_-Induced Mitochondrial Dynamics Disturbance

Mechanically, whether and how the increased expression of ac-p53 contributed to ISO/H_2_O_2_-evoked mitochondrial dynamics disturbance remains unclear. More recently, a study from Kong et al. described that p-p53 (ser15) interacts with Bak and PHB1 and involves in the regulation of mitochondrial dynamics in gynecologic cancer cells [18]. Therefore, we hypothesized that ac-p53 might interact with PHBs and facilitate the dissociation of the OPA1-PHBs complex, ultimately resulting in L-OPA1 proteolysis. To investigate this mechanism, we first analyzed the mRNA expression of Opa1 in NRCMs subjected to different interventions and found that Opa1 mRNA level was not significantly changed in ISO or H_2_O_2_-treated NRCMs compared to the control group, as shown in Figure 5A,B, suggesting a posttranscriptional regulation of L-OPA1 processing. 

Next, we examined the interaction between PHBs and ac-p53 by exposing NRCMs to 10 μM ISO for 24 h and analyzing the levels of L-OPA1, PHB2, PHB1, and ac-p53 in whole-cell lysates. Our results showed that ISO treatment increased the protein levels of ac-p53 while downregulating the expression of L-OPA1, PHB1, and PHB2 proteins. Immunoprecipitation assays further revealed that ISO treatment significantly enhanced the interaction between ac-p53 and PHB2 rather than PHB1 (Figure 6A,B). 

We further investigated whether ac-p53 facilitates the dissociation of OPA1 from the OPA1-PHB2 complex in response to ISO. A mutant p53 (K371R) was constructed by replacing a lysine (K) with arginine (R) at the 371 sites of wild-type p53, mimicking deacetylated p53, to examine the alteration of the interactions between PHB2 and OPA1 or ac-p53 by PHB2 immunoprecipitates. Our results showed that the p53 (K371R) mutation produced differential effects on the interactions between PHB2 and ac-p53 or OPA1. Specifically, we observed a marked reduction in the former interaction but a distinct enhancement in the latter (Figure 7A). Moreover, ISO or H_2_O_2_ treatment led to a significant increase in ac-p53 expression but a marked decrease in PHB2 expression only in the cells transfected with wild-type p53, not the K371R mutant, whereas total p53 levels remained unchanged following ISO or H_2_O_2_ exposure regardless of both types of plasmid-transfected cells (Figure 7B). 

To reinforce this hypothesis, we performed an immunostaining experiment to further investigate the impacts of ac-p53 on the interaction of OPA1 and PHB2 in mitochondria in NRCMs. The results of this study demonstrated that ISO or H_2_O_2_ treatment resulted in a decrease in the colocalization of OPA1 and PHB2 within NRCM mitochondria, whereas FL pretreatment significantly enhanced their colocalization. Expectedly, we observed that PFT-α treatment produced a similar effect on the colocalization of OPA1 and PHB2 to FL, while EX527 abrogated the enhanced colocalization of OPA1 and PHB2 induced by FL treatment in ISO or H_2_O_2_-challenged cells as demonstrated in Figure 8A,B. Taken together, these findings suggest a novel mechanism in which p53 acetylation within mitochondria suppresses the formation of the OPA1-PHB2 complex by interacting with PHB2, thereby facilitating L-OPA1 processing and ultimately leading to mitochondrial dynamics imbalance in the ISO or H_2_O_2_-stimulated cardiomyocytes. Activation of Flt3 enhances the stability of the OPA1-PHB2 complex by increasing SIRT1-dependent p53 deacetylation, thus improving mitochondrial dynamics disturbance induced by ISO or H_2_O_2_ in cardiac myocytes.

### 3.4. Flt3 Activation Ameliorates LAD-Induced Cardiac Remodeling 

Based on the cellular results above, we conducted further investigation into the hypothesis of this study in ischemic cardiac remodeling in vivo. Firstly, we examined the impact of Flt3 activation on LAD-induced cardiac remodeling in mice. First, we found that FL administration significantly attenuated the arched ST segment elevations induced by LAD in ECG recordings of mice. Our echocardiographic analysis revealed that the mice in the LAD group exhibited significant increases in LVIDd, LVIDs, LVEDV, and LVESV but distinct decreases in LVEF% and LVFS%. However, the FL administration evidently reversed these changes, as shown in Figure 9. Moreover, FL intervention markedly ameliorated the cardiac infarct area from 39% in the MI group to 12.8% in the MI + FL group and dramatically reduced the area of post-infarct fibrosis from 14.9 to 6.1%, as depicted in Figure 10. These results indicate that activation of Flt3 by FL efficiently mitigated LAD-induced cardiac remodeling.

### 3.5. Flt3 Activation Reduced L-OPA1 Processing by Hindering the Interaction between p53 and PHBs in Mitochondria in LAD- or ISO-Induced Cardiac Remodeling

We next investigated whether p53 acetylation affected the interaction between PHBs and OPA1 in LAD-induced cardiac remodeling. As shown in Figure 11, the immunoblotting analysis demonstrated that the protein expression of L-OPA1, SIRT1, PHB2, and PHB1, as well as the ratio of L-OPA1 to S-OPA1 was significantly reduced with increased ac-p53 and p53 in the MI group. However, the FL administration obviously reversed the alterations. We conducted IP-PHB2 analysis on myocardial mitochondrial samples from the infarct border zone to investigate the interaction between PHB2 and ac-p53 or OPA1 in mitochondria. Our results revealed that MI led to a significant increase in the interaction between ac-p53 and PHB2 while causing an obvious decrease in the interaction between OPA1 and PHB2 in the mitochondria of cardiomyocytes. However, FL administration overtly attenuated these alterations, implicating that Flt3 activation inhibits mitochondrial L-OPA1 processing and subsequent mitochondrial dynamic imbalance through SIRT1/p53 pathway, which may contribute to LAD-induced cardiac remodeling. This mechanism was further strengthened in the setting of ISO-induced cardiac remodeling in vivo, as evidenced by the fact that FL intervention significantly mitigated the reduction of mitochondrial L-OPA1, PHB1, and PHB2 protein expression, accompanied by elevated ac-p53 levels in mitochondria.

## 4. Discussion

In the present study, we provide further evidence that activation of Flt3 plays a crucial role in improving maladaptive cardiac remodeling by restoring mitochondrial dynamics balance through the SIRT1/p53 pathway. Specifically, we demonstrate, for the first time, that acetylated p53 levels are significantly increased in the mitochondria of cardiomyocytes under cardiac stress conditions, specifically in cases of ISO- or LAD-induced pathological cardiac remodeling. This increase may lead to a reduction in the formation of the OPA1-PHB2 complex by binding with PHB2, thereby promoting L-OPA1 processing and subsequent disturbances of mitochondrial dynamics. Flt3 activation reduces L-OPA1 processing by impeding the interaction between ac-p53 and PHB2 in mitochondria via the SIRT1 pathway, thereby ameliorating mitochondrial dynamics disturbances, which probably contributes to mitigating ISO- or LAD-induced maladaptive cardiac remodeling. 

Cardiac remodeling-related pathological mechanisms involve multiple factors, including inflammation, oxidative stress, cell death, and fibrosis [19]. The heart is one of the most energy-demanding organs in the human body, and the function and homeostasis of the heart rely on mitochondrial oxidative metabolism to synthesize ATP for proper function [20]. Mitochondria consist of a double membrane that regulates their dynamics through fission and fusion events, which impact the morphological structure and functionality. Undoubtedly, exploring innovative approaches that target mitochondrial remodeling is crucial for the treatment of cardiac remodeling, given the close association between an imbalance in mitochondrial fission and fusion. Flt3, a receptor tyrosine kinase, mediates cell survival through its activation by its specific ligand FL. Recent evidence from Pfister et al. and our previous study has revealed that Flt3 activation efficiently ameliorates pathological cardiac hypertrophy and remodeling induced not only by ischemic insult but also by pressure overload of angiotensin II or β-adrenergic stimulation [13,14,15]. In particular, our previously published study demonstrated that Flt3 activation restored the balancing of mitochondrial dynamics in ISO-induced cardiac hypertrophy via the SIRT1/p53 pathways, resulting in increased expression of Drp1 protein and involvement in mitochondrial fission. In this study, we further highlight that Flt3 activation mitigates mitochondrial fragmentation by sequestering L-OPA1 and protecting it from proteolysis.

Mitochondria dynamics are meticulously regulated by several key proteins, including the fusion proteins OPA1 and Mfn2, as well as the fission proteins DRP1 and Fis1 [21]. We observed that gene expression levels for Mfn2, Drp1, and Fis1 were significantly altered in response to the different interventions, as indicated in our study. However, the mRNA expression of OPA1 did not change significantly; instead, its protein levels changed. OPA1 is an inner mitochondrial membrane (IMM) protein that anchors to the IMM through its transmembrane domain. In the heart, there are five different isoforms of OPA1, and L-OPA1 isoforms are necessary for inner membrane fusion and maintaining inner membrane structure. However, L-OPA1 can be cleaved by proteases OMA1 and YMEL1 at two specific cleavage sites (S1 and S2), generating three S-OPA1 isoforms. Excessive cleavage of OPA1 by activated OMA1 in response to stress or damage, such as oxidative stress and ischemia, and an imbalance between L-OPA1 and S-OPA1 would lead to mitochondrial fragmentation and compromised mitochondrial function [4]. Therefore, the expression of OPA1 protein decreased, rather than the mRNA expression, in response to ISO, H_2_O_2,_ or LAD ischemia challenges, and Flt3 activation might involve an additional mechanism, such as L-OPA1 proteolytic degradation. We found that the increased expression of ac-p53 proteins and the decreased PHB2 in mitochondria of cardiac myocytes or tissues was closely associated with the reduction of L-OPA1 protein in the settings of this study in vitro or in vivo. 

p53 proteins play a crucial role in regulating mitochondrial remodeling by controlling mitochondrial content, fusion/fission processes, and intracellular signaling molecules associated with apoptosis pathways [22]. Recently, Guo et al. demonstrated that DRP1 stabilized p53 through the interaction of Drp1′s GTPase domain and p53′s DNA binding domain. Furthermore, DRP1 is necessary for p53 translocation to mitochondria to trigger neuron necrosis under oxidative stress [23]. Mukherjee et al. used immunoprecipitation and proximity ligation assays to demonstrate that DRP1 and p53 interaction through direct protein–protein binding mediated p53 mitochondrial localization and dysfunction in LPS-induced septic cardiomyopathy [24]. Our previous study has demonstrated that the protein expression of DRP1 and p53 is significantly increased in NRCMs exposed to ISO, while Flt3 activation reverses their alterations [13]. Similarly, in the model of this study, the interaction between p53 and DRP1 likely plays an important role in regulating mitochondrial dynamics in response to ISO or H_2_O_2_ as well as Flt3 activation, which warrants further investigation. While these studies highlight the critical role of p53 in the outer membrane cleavage, it remains limited understanding regarding whether p53 also plays a significant role in the inner membrane cleavage during mitochondrial fragmentation. Our present study provides new evidence supporting a viewpoint that ac-p53 interacts with PHB2 in mitochondria, leading to destabilization or dissociation of the PHB2-L-OPA1 complex under conditions of cardiac stress or damage. This results in proteolytic degradation of L-OPA1 and fragmentation of the mitochondrial network. Conversely, Flt3 activation reduces ac-p53 expression levels in mitochondria through SIRT1, rescuing the stability of the PHB2-L-OPA1 complex and restoring mitochondrial network integrity.

PHB proteins are highly expressed in cardiomyocytes that require high energy, making them more susceptible to mitochondrial dysfunction. Human PHB2 possesses an uncleavable mitochondrial targeting sequence at the N-terminus [25]. By contrast, while the N-terminus of PHB1 is necessary for mitochondrial localization, it does not contain a typical mitochondrial targeting sequence. Functionally, PHBs have been implicated in the stability of mitochondrial morphology and the regulation of mitochondrial dynamics via various mechanisms. For example, protease OMA1 is commonly sequestered within the ring complex formed by PHB1 and PHB2; however, upon cell stress, disruption of the PHBs complex would lead to OMA1 release [9]. A recent study showed that Bif-1, also known as endophilin B1 and serving as a Bax interacting protein, translocated to the mitochondria where it bound with PHB2, leading to disruption of the PHBs complex and proteolytic degradation of L-OPA1 during renal ischemia/reperfusion injury [26]. Of note, Kong et al. found that phosphorylated p53 (Ser15) interacts with PHB1 in chemosensitive cervical cancer cells exposed to cisplatin, resulting in dissociation of the OPA1-PHB1 complex [18,27]. In the present study, we confirm that ac-p53 promotes the dissociation of the OPA1-PHBs complex by binding to PHB2 during cardiomyocyte stress induced by ISO, H_2_O_2,_ or LAD infraction, since the p53 K371R mutant, which lacks an acetylated site, fails to bind to PHB2, thereby impeding the formation of ac-p53-PHB2 complex and stabilizing the OPA1-PHBs complex under cellular stress. In contrast, activation of Flt3 enhanced the SIRT1 protein expression and reduced ac-p53 expression, resulting in similar effects on the OPA1-PHBs complex and OPA1 processing in response to challenges from ISO, H_2_O_2,_ or LAD infraction. This provides a protective effect against mitochondrial network fragmentation in cardiac maladaptive remodeling. These studies indicate that p53 can translocate to mitochondria in the form of either phospho-p53 or ac-p53 and plays a critical role in regulating mitochondrial fission or fusion at the outer or inner mitochondrial membrane. 

## 5. Conclusions

In conclusion, our study indicates that Flt3 activation mitigates mitochondrial dynamics imbalance and dysfunction during cardiac maladaptive remodeling by hindering the interaction between ac-p53 and PHB2 in mitochondria, which leads to a reduction in L-OPA1 processing. These findings suggest that Flt3 may be a promising therapeutic target for cardiac remodeling and heart failure since protecting L-OPA1 from processing can efficiently reduce mitochondrial network fragmentation and apoptosis.

## Figures and Tables

**Figure 1 antioxidants-12-01657-f001:**
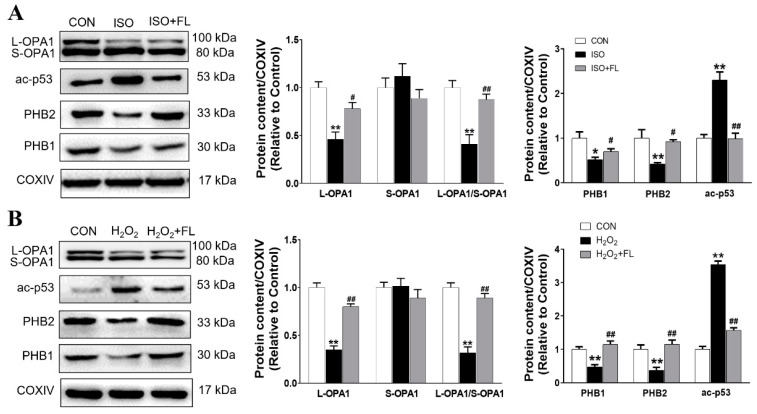
Effects of Flt3 activation on the protein expression of L-OPA1, S-OPA1, ac-p53, and PHBs in the mitochondria of ISO or H_2_O_2_-treated NRCMs in vitro. (**A**,**B**) Representative western blots showing the effects of FL treatment on the expression of L-OPA1, S-OPA1, ac-p53, PHB1, and PHB2 in the mitochondria of ISO or H_2_O_2_-treated NRCMs, and the quantitative analysis (n = 3 independent experiments). Data were expressed as mean ± SD, * *p* < 0.05, ** *p* < 0.01 vs. CON, **^#^**
*p* < 0.05, **^##^**
*p* < 0.01 vs. ISO/H_2_O_2_.

**Figure 2 antioxidants-12-01657-f002:**
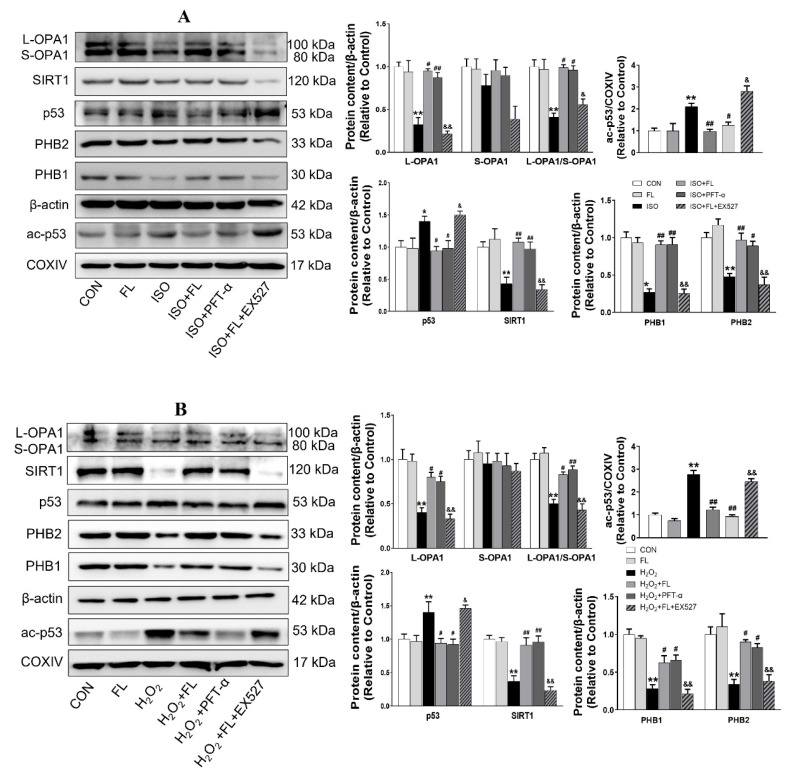
Flt3 activation decreases L-OPA1 processing via SIRT1/p53 pathway in vitro. (**A**,**B**) Representative western blots showing the effects of FL treatment, p53 inhibitor PFT-α, and SIRT1 inhibitor EX527 on the expression of L-OPA1, S-OPA1, SIRT1, p53, ac-p53, PHB1, and PHB2 in ISO or H_2_O_2_-treated NRCMs, and the quantitative analysis (n = 3 independent experiments). β-actin was used as an internal reference of cytoplasmic proteins, while COXIV was that of mitochondrial lysate proteins. Data were expressed as mean ± SD, * *p* < 0.05, ** *p* < 0.01 vs. CON, **^#^**
*p* < 0.05, **^##^**
*p* < 0.01 vs. ISO/H_2_O_2_, ^&^
*p* < 0.05, ^&&^
*p* <0.01 vs. ISO/H_2_O_2_ + FL.

**Figure 3 antioxidants-12-01657-f003:**
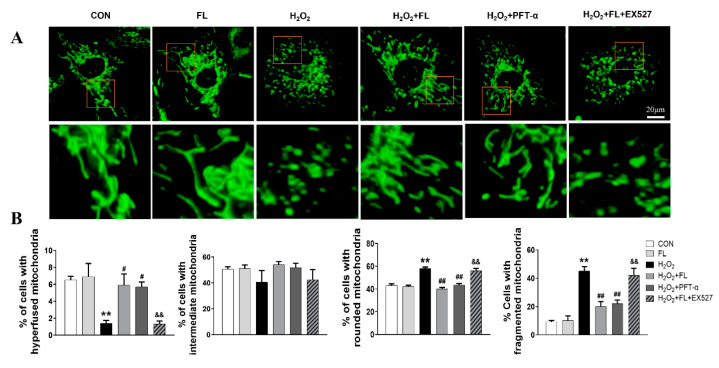
Flt3 activation restores ISO- or H_2_O_2_-induced mitochondrial dynamics imbalance by reducing L-OPA1 processing in vitro (**A**) Representative fluorescence images of mitochondrial morphology of NRCMs challenged by H_2_O_2_, FL, or in combined with PFT-α or EX527 intervention by using Mito-Tracker Green probe (the scale, 20 µm). The figure below shows a zoomed-in view of the red boxed region. (**B**) Quantitative analysis of fragmented, hyperfused (at least one mitochondrion > 5 µm in length), intermediate (at least one mitochondrion between 5 to 2 µm but none more the 5 µm in length), and rounded (none longer than 2 µm) mitochondria (n = 125 cells/group). Data were expressed as mean ± SD, ** *p* < 0.01 vs. CON, ^#^
*p* < 0.05, ^##^
*p* < 0.01 vs. ISO/H_2_O_2_, ^&&^
*p* < 0.01 vs. ISO/H_2_O_2_ + FL.

**Figure 4 antioxidants-12-01657-f004:**
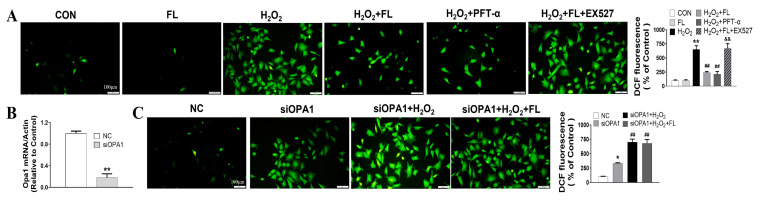
Flt3 activation inhibits H_2_O_2_-induced ROS and apoptosis by improving mitochondrial dynamics disturbance in vitro. (**A**) Representative photographs and quantification of ROS by DCFH-DA probe in NRCMs pretreated with FL, PFT-α, and EX527 followed by exposure to H_2_O_2_, n = 10 random fields (Scale bar, 100 μm). (**B**) Quantitative data of the Opa1 mRNA expression after treatment with OPA1 siRNA or to negative control (NC). (**C**) Representative photographs and quantification of ROS by DCFH-DA probe in NRCMs pretreated with FL and siOPA1 followed by exposure to H_2_O_2_, n = 10 random fields (Scale bar, 100 μm). (**D**,**E**) Representative photographs and quantification of apoptosis by Annexin V-FITC/PI probe in NRCMs exposed to H_2_O_2_ and performed subsequent drug intervention, n = 50 individual cells (Scale bar, 500 μm). Data were expressed as mean ± SD, * *p* < 0.05, ** *p* < 0.01 vs. CON/NC, ^**##**^
*p* < 0.01 vs. ISO/H_2_O_2_/siOPA1, ^&&^
*p* < 0.01 vs. ISO/H_2_O_2_ + FL.

**Figure 5 antioxidants-12-01657-f005:**
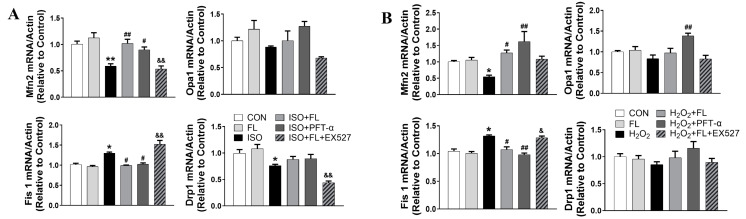
The mRNA expression of mitochondrial fusion and fission genes in ISO- or H_2_O_2_-induced mitochondrial dynamics dysfunction in NRCMs. Quantitative data of the expression of mitochondrial fission and fusion-related genes (Mfn2, Opa1, Fis1, Drp1) in the different groups of NRCMs evoked by ISO (**A**) or H_2_O_2_ (**B**) (n = 3 independent experiments). Data were expressed as mean ± SD, * *p* < 0.05, ** *p* < 0.01 vs. CON, ^#^
*p* < 0.05, ^##^
*p* < 0.01 vs. ISO/H_2_O_2_, ^&^
*p* < 0.05, ^&&^
*p* < 0.01 vs. ISO/H_2_O_2_ + FL.

**Figure 6 antioxidants-12-01657-f006:**
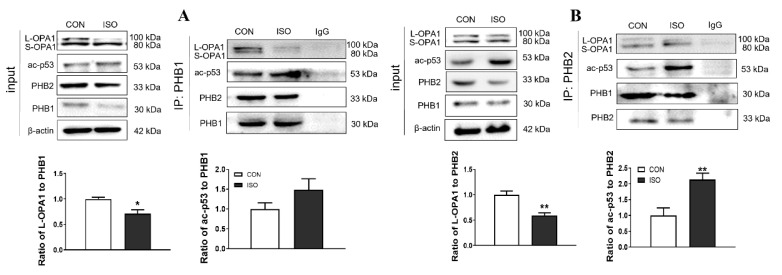
The interaction between PHBs and ac-p53 in ISO-stimulated NRCMs. (**A**,**B**) Representative western blot images showing the expression of OPA1, ac-p53, PHB1, and PHB2 in ISO-treated NRCMs. Cell lysates were immunoprecipitated with IgG (control; lane 3) or PHB1/PHB2 antibody. Protein-protein interaction was determined by western immunoprecipitation. Representative western blot images and the quantitative analysis showed that ISO significantly increased the interaction of ac-p53 and PHB2 but not PHB1 in NRCMs (n = 3 independent experiments). Data were expressed as mean ± SD, * *p* < 0.05, ** *p* < 0.01 vs. CON.

**Figure 7 antioxidants-12-01657-f007:**
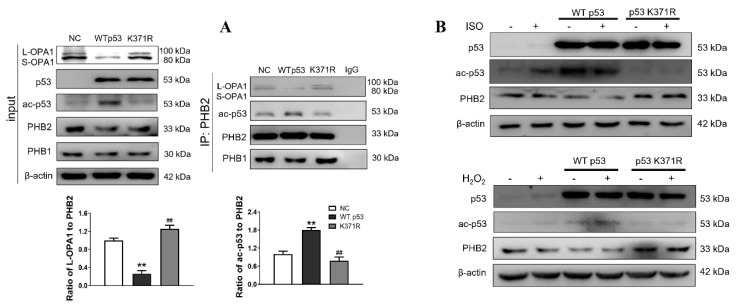
Acetylated p53 is necessary for the ISO- or H_2_O_2_-induced dissociation of OPA1 from the OPA1-PHB2 complex in cardiomyocytes. (**A**) H9c2 cells were transfected with wild-type p53 (WTp53), lysine 371 mutant p53 (p53 K371R) plasmid. The overexpression was confirmed by the p53 antibody in the whole cell lysates. Protein−protein interaction was determined by western immunoprecipitation. Representative western blot images and the quantitative analysis showed that p53 K371R significantly enhanced the interaction of OPA1 and PHB2 and decreased that of PHB2 and ac-p53 (n = 3 independent experiments). (**B**) Representative western blots for the expression of p53 K371R plasmids. Plasmids were transfected into H9c2 cells and then treated with ISO or H_2_O_2_ for testing the changes of p53, ac-p53, and PHB2. Data were expressed as mean ± SD, ** *p* < 0.01 vs. NC, ^##^
*p* < 0.01 vs. WTp53.

**Figure 8 antioxidants-12-01657-f008:**
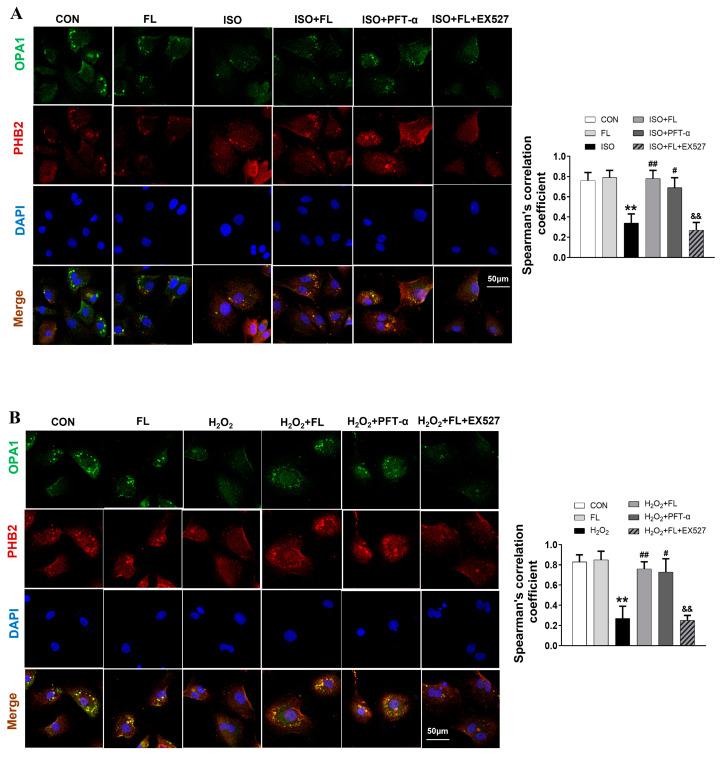
Flt3 activation enhances ISO- or H_2_O_2_-induced reduction of OPA1 and PHB2 colocalization within NRCM mitochondria. (**A**,**B**) Cultured NRCMs in the different groups were contained with anti-OPA1 antibody (green), anti-PHB2 antibody (red), and DAPI (Scale bar, 50 μm). Representative fluorescence images were shown, and the colocalization of OPA1-PHB2 based on Spearman’s correlation in the merged images was analyzed (n = 30 cells from 3 independent experiments). Data were expressed as mean ± SD, ** *p* < 0.01 vs. CON, ^#^
*p* < 0.05, ^##^
*p* < 0.01 vs. ISO/H_2_O_2_, ^&&^
*p* < 0.01 vs. ISO/H_2_O_2_ + FL.

**Figure 9 antioxidants-12-01657-f009:**
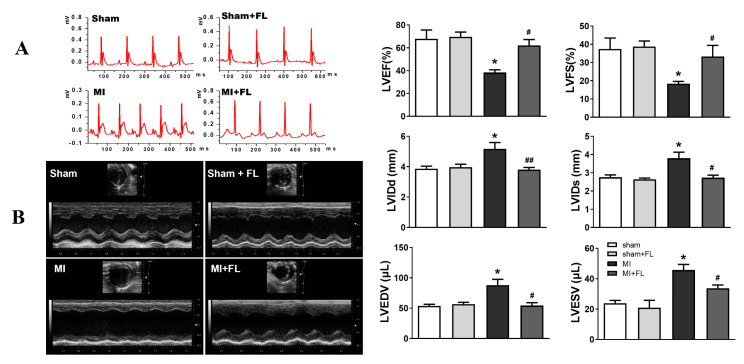
Flt3 activation improves the cardiac dysfunction of mice induced by LAD. (**A**) Representative images of electrocardiographic (ECG) recordings in the four groups at 28 days post-MI (n = 5). (**B**) Representative M-mode echocardiography images of mice in the different groups and bar graphs showing quantification of left ventricular internal diastolic diameter (LVIDd), left ventricular internal systolic diameter (LVIDs), left ventricular ejection fraction (LVEF%), left ventricular fractional shortening (LVFS%), left ventricular end-diastolic volume (LVEDV) and left ventricular end-systolic volume (LVESV) (n = 5). Data were expressed as mean ± SD, * *p* < 0.05 vs. sham, **^#^**
*p* < 0.05, ^**##**^
*p* < 0.01 vs. MI.

**Figure 10 antioxidants-12-01657-f010:**
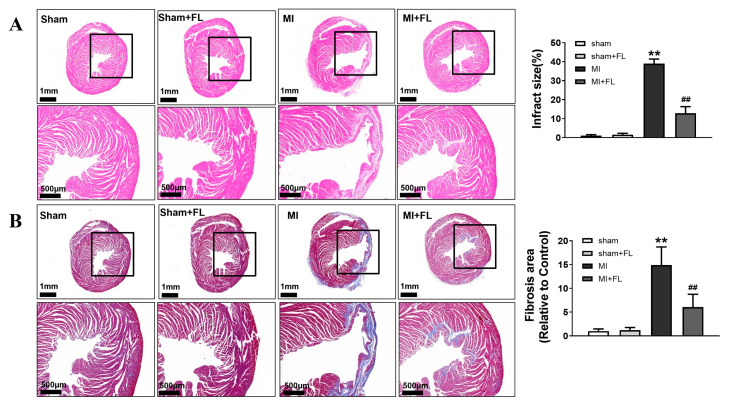
Flt3 activation ameliorates LAD-induced cardiac remodeling in mice. (**A**) Representative images of hematoxylin and eosin staining of the transection of the mouse hearts showing the gross cardiac morphology (Scale bar, 1 mm in the upper, 500 μm in the bottom) and quantitative analysis of relative infarct size (n = 5). (**B**) Representative images and quantitative analysis of Masson’s trichrome intensity calculated from the Masson-stained heart (n = 5). The figures below show the zoomed-in view of the black boxed region. Data were expressed as mean ± SD, ** *p* < 0.01 vs. sham, **^##^**
*p* < 0.01 vs. MI.

**Figure 11 antioxidants-12-01657-f011:**
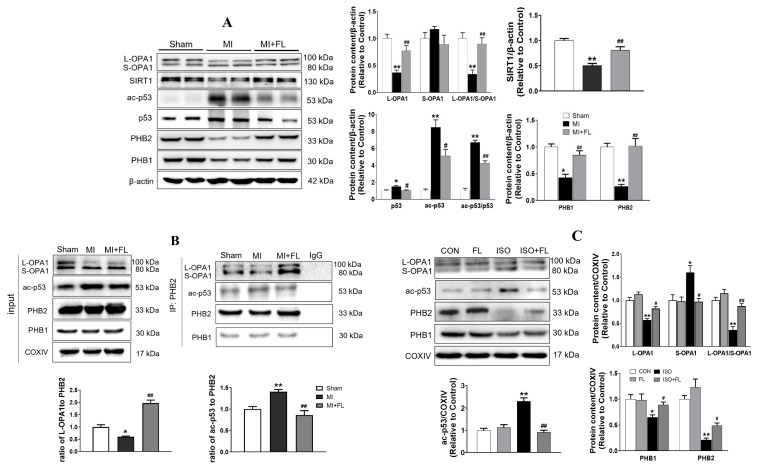
Flt3 activation reduces L-OPA1 processing by hindering interaction between p53 and
PHBs in mitochondria in LAD- or ISO-induced cardiac remodeling in vivo. (**A**)
Representative western blots and quantitative analysis of L-OPA1, S-OPA1, 
SIRT1, p53, ac-p53, PHB1, and PHB2 in heart tissues of mice (n = 3 independent 
experiments). (**B**) Representative western blot images showing the 
expression of OPA1, ac-p53, PHB1, and PHB2 in mitochondria of MI and MI + FL 
mouse hearts. Mitochondria fraction was separated and immunoprecipitated with 
IgG (control; lane 4) or PHB2 antibody. Protein–protein interaction was 
determined by western immunoprecipitation. Representative western blot images 
and the quantitative analysis showing that FL significantly decreased the interaction 
of ac-p53 and PHB2 and increased the interaction of OPA1 and PHB2 in 
mitochondria compared with MI (n = 3 independent experiments). (**C**) 
Representative western blot images showing the expression of OPA1, ac-p53, PHB1, 
and PHB2 in mitochondria in ISO and ISO + FL mouse hearts. Data were expressed 
as mean ± SD, * *p* < 0.05, ** *p* < 0.01 vs. 
sham/CON, ^#^
*p* < 0.05, ^##^
*p* < 0.01 vs. 
MI/ISO.

**Table 1 antioxidants-12-01657-t001:** Primers used for quantitative polymerase chain reaction.

Primers	Forward	Reverse
*Opa1*	CCGAAAGCCTCAGCTTGTTG	GCAGAAGTTCTTCCTGAAGTTGG
*Mfn2*	GTGACGTGTTGGGTGTGAT	GGACATCTCGTTTCTAGCTGGT
*Drp1* *Fis 1*	CGAAAACTGTCTGCCCGAGA CCAGAACAACCAGGCCAAGGAG	GCATTACTGCCTTTGGGACG CACAGCCAGTCCAATGAGTCCAG
*Actin*	TGTCACCAACTGGGACGATA	GGGGTGTTGAAGGTCTCAAA

## Data Availability

The data supporting this research are included in the manuscript.

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
