# Peer review of "Flt3 Activation Mitigates Mitochondrial Fragmentation and Heart Dysfunction through Rebalanced L-OPA1 Processing by Hindering the Interaction between Acetylated p53 and PHB2 in Cardiac Remodeling"

_antioxidants, 2023, doi:10.3390/antiox12091657_

Round 1

Reviewer 1 Report

In this manuscript the authors present a huge amount of data and experiments but, unfortunately, the take at home message is not so clear. Two different challenges (ISO and H2O2) are presented in parallel, which is both very interesting but sometimes a little confusing. The authors described a novel mechanism by which Flt3 impacts mitochondrial fragmentation, and notably L-OPA1. This manuscript is interesting, but I have some concerns that need to be addressed as follows:

Major comments:

1.      Abstract : The results part of this abstract is quite difficult to clearly understand, mainly due to a large number of abbreviations which are not necessarily explained (e.g. PHB2, L-OPA1). The authors could simplify certain sentences of the abstract in order to improve the general understanding of the manuscript.

2.      Methods 2.8 : The authors should explicit how much photographs and cells are used for quantification, as it was well explain for the 2.7 paragraph

3.      Methods 2.14 : regarding the statistical part, maybe it will be better to represent data as boxplot, because n=3 is not sufficient to perform parametric test such as Student’s t-test. I will be better to use non parametric test such as Mann and Whitney at least for western blot and RTqPCR data.

4.      The authors present a huge amount of data, particularly two different challenges (ISO and H2O2). The link between this two treatment is a little lacking. Why the authors are interesting by both ? Is it because both induce oxidative stress or mitochondrial dysfunction ? Is it because both induce the same signalling pathway or on the contrary different signalling pathway ? Explaining the challenge a little more at the beginning would greatly improve the overall understanding of the manuscript.

5.      Figure 1 : It will be interesting to show the efficiency of treatments in the same figure. Does Iso well induce hypertrophy ? I assume yes, as it well described in the previous study of the authors (and classically used in literature). Does H2O2 well induce oxidative stress ? Yes, but the answer only appears in Figure 4. Maybe this Figure gained to be presented together at least for the 3 groups : control, treated (ISO or H2O2) and treated + FL. As the authors described mitochondrial effect of Flt3, it will be interesting to specifically quantify mitochondrial oxidative stress (with MitoSOX for example) .It will be also interesting to have total p53 and ac-p53 / p53 quantification, as described for the in vivo part.

6.      Figure 2 : Could the authors explained why quantification are made sometimes on actin and sometimes on COXIV ? Is it because of the subcellular location of the proteins?

7.      Figure 4 : The siRNA of OPA1 increased the complexicity of data. It will be easier if data are separated in two panels A for treatment and B for siRNA. Moreover, the authors should add RNA or protein quantification of inhibition of OPA by siRNA (maybe in Figure 5).

8.      Figure 6: Interaction experiments are interesting but maybe needed to be support these data by immunostaining (or even proximity ligation assay) if it’s technically feasible, to confirm if it’s interaction is in mitochondria.

9.      Figure 7: Why the authors use cardiomyoblasts H9c2 for transfection ? I assumed that NRVM are very difficult to transfect but this point need to be discussed.

10.  Figure 8: It will be a great benefit to add mitochondrial staining (such as Mito Tracker deep red) in colocalization staining or maybe PLA to confirm if the colocaliztion is in in mitochondria.

11.  General comment for discussion: There is another sirtuin (SIRT3) known to be localized in mitochondria and modulated during cardiac hypertrophy. What is known about a link between p53, Flt3 and SIRT3 ? Does this sirtuin also be involved in the pathway described by the authors ?

Minor comments:

1.                  Line 17 : Lack of empty space between H2O2 and in vitro

2.                  Line 20 : I assumed authors would say “in mitochondria” instead of “in mitochondrial”

3.                  Table 1 the primer sequence for Fis1 should be in one line if possible

4.                  It seems that Figure 1 legend is on two pages (begin line 277-278 and end line 297-300).

5.                  Line 282 PFT-α is in higher characters that the rest of the legend

6.                  Line 575: I assumed authors would say “phospho-p53” instead of “phosph-p53”

Reviewer 2 Report

According to the abstract, this study by Zhang et al investigate the involvement of p53-regulated optic atrophy 1 (OPA1) processing and mitochondrial fragmentation in Flt3-mediated improvement of cardiac remodeling. The introductions specifies and states that “this study aims to investigate whether the regulation of L-OPA1 processing by the ac-p53-PHB2 complex in cardiomyocytes represents a universal mechanism underlying mitochondrial dynamics imbalance in ISO- and LAD-induced cardiac remodeling, and if also explores the potential of Flt3 to target this mechanism and improve mitochondrial dynamics in this context.

I am not an expert on mitochondrial fission and fusion, but on mitochondrial bioenergetics and development. So, my comments represent more the view point of someone knowing mitochondria, but not necessarily all the details of mitochondrial fission and fusion. I have more problems to understand how the provided background information lead to the aim of this study than understanding the actual results. Therefore I would like to ask the authors to improve the introduction:

What is the role of PHB1 and PHB2? The referenced paper 8 does not identify either what PHB1 or PHB2 are. In addition, the referenced paper 8 is not a research paper, but a commentary to 2 papers (Head et al 2009, Ehses et al 2009). Understanding the role of PHB1 and PHB2 becomes important, because of the data presented in figure 6. Please provide definitions and maybe a research paper for given context.  

Flt3: not introduced; what Flt3 is, what is its physiological role? How is the function of Flt3 related to mitochondria or fission and fusion? What is Flt3-ligand? Please add some relevant background information into the introduction.

I have problems with the term mitochondrial dynamic imbalance. When becomes mitochondrial dynamic a dynamic imbalance? Since imbalance with respect to this paper include both, too much fission or too much fusion, the authors should specify, what is meant. Similarly, the authors write frequently FL intervention, which also could include activation and inhibition, but this manuscript uses only an activating Flt3-ligand.

Line 60: Typo act instead of acts

Line 77: typo if instead of it

Methods:

Line 110/111: For how long were the cells cultured before switching to serum-free medium?

Line 113: What is CON?

Results:

The results show a huge number of Western Blots designed to provide evidence that Flt3 is an important factor that somehow regulates through a SIRT1/p53 pathway.  

Line 265: Typo; their instead of there

Line 277: Please add A and B for figure 1A and 1B into the legend.

Line 281: Please add A and B for figure 2A and 2B into the legend

Line 297-300: This paragraph appears to be out of place.

Line 307: typo shown instead of showed

Figure 4A and 4B: Why are the graphs analyzing the images divided? Legends in graph are too small

Figure 4B: Please provide better images; cells are not visible.

Figure 5: legend on the graph are too small

Figure 8: Figure legend mentions Spearman correlation, but graph show Pearson coefficient. Which one is it?  

Figure 10: A and B show images of the same heart. This figure is not linked to the description of the results. Please focus for example by a boxed area, how damaged and healthy tissue looks.  

Moderate editing of English language required. There are some grammatical errors.

Round 2

Reviewer 1 Report

I would like to thank the authors for their convincing responses to all my previous comments. I have just one last minor suggestion about the introduction. Concerning the part added by the authors on ISO and LAD (see line 81-84), I find it a little too short and I would appreciated it to be as detailed as in the response to the reviewers (previous comment n4).

And line 57, there is a "e" between membrane and and that need to be remove.
